# COVID and Gender: A Narrative Review of the Asia-Pacific Region

**DOI:** 10.3390/ijerph20010245

**Published:** 2022-12-23

**Authors:** Colin W. Binns, Mi Kyung Lee, Thi Thuy Duong Doan, Andy Lee, Minh Pham, Yun Zhao

**Affiliations:** 1School of Public Health, Curtin University, Bentley 6102, Australia; 2Public Heath Nutrition Consultant, Perth 6845, Australia; 3Department of Population and Reproductive Health, Faculty of Social Sciences, Behavior and Health Education, Hanoi University of Public Health, 1A Duc Thang Street, Bac Tu Liem District, Hanoi 10000, Vietnam; 4Department of Preventive Medicine, Thai Nguyen University of Medicine and Pharmacy, Thai Nguyen 250000, Vietnam

**Keywords:** COVID-19, gender, women’s health, pandemic, long COVID, Asia

## Abstract

The COVID-19 pandemic has been the largest infectious disease epidemic to affect the human race since the great influenza pandemic of 1918-19 and is close to approaching the number of deaths from the earlier epidemic. A review of available data and the numerous currently available studies on COVID-19 shows that the rate of clinical cases is about 10% greater in females than males in Asia. However, the number of deaths is greater in males than in females. Women are more likely to experience the psychological effects of COVID-19 during and after acute infections. A significant proportion of acute COVID-19 infections continue and their prolonged symptoms have been reported. Further studies are needed, including detailed serology, to measure and monitor the incidence of COVID-19. The pandemic has had a widespread impact on broader societies including shortages of food, lockdowns and isolation. The number of orphans in developing countries has increased. Women have had to bear the major impacts of these community effects. More research is required to develop better vaccines acting against new strains of the virus and to develop systems to distribute vaccines to all people.

## 1. Introduction

The COVID-19 pandemic has been the largest infectious disease epidemic to affect the human race since the Great Influenza Pandemic of 1918-19. Estimates of the number of deaths in the influenza epidemic range as high as 100 million, but a figure of 50 million is more probable [1]. As of November 2022, the number of global deaths due to COVID-19 that have been recorded is 6.6 million [2]. 

The deaths reported to WHO are very likely to be underestimated, as many cases of COVID-19-related deaths go unreported and this can be assessed by monitoring excess mortality. In the pandemic up to the end of 2021 WHO had recorded 5.4 million deaths and estimated that excess mortality during this period accounted for a further 9.5 million deaths. This brings the total number of deaths associated with COVID-19 to 14.9 million [3]. Extrapolating the data to November 2022 suggests that the true number of deaths is likely to be closer to 18.1 million. The advent of vaccinations against COVID-19 is estimated to have averted approximately 20 million deaths by the end of 2021 [4]. If the actual number of deaths and the number of deaths averted are considered together, the total is comparable to the Great Influenza Epidemic. When the experience of 2022 is included, COVID-19 may be the greatest epidemic of all time. The WHO estimates for excess mortality in the South East Asia and Western Pacific regions were 5.99 (40.2% of global excess mortality) and 0.12 million, respectively [3].

The pre-COVID-19 world was one in which gender equality was not yet the norm. Inequality has been worsened by the pandemic. Discriminatory patterns exist in terms of access to, ownership and control of productive resources Relative income poverty, physical vulnerability, and lack of fully equitable and meaningful participation at all levels of decision-making processes have worsened. This is particularly the case for minority groups and indigenous women in Asia [5].

### 1.1. Definitions and Syndrome Descriptions

Defining COVID-19 disease clinically is difficult because of the wide spectrum of symptoms, signs and outcomes. In the early stages of the epidemic, numerous publications described symptoms related to COVID-19 infections. The following symptoms of COVID-19 and long COVID-19 are listed by the US CDC [6]: 

People with COVID-19 have reported a wide range of symptoms – ranging from mild symptoms to severe illness. Symptoms may appear 2–14 days after exposure to the virus. Symptoms may include one or more of the following. Fever, cough, dyspnoea, fatigue, muscle or body aches, headache, loss of taste or smell (anosmia), sore throat, nasal discharge, nausea, vomiting or diarrhoea. It is all too obvious that these symptoms overlap with a host of other illnesses, mostly of viral aetiology. For this reason, little progress was made with the epidemiology of COVID-19 until the advent of various types of serological tests. 

The WHO classifies COVID-19 cases into three categories [7]:

Suspected: based on clinical or epidemiological criteria;

Probable: A patient who meets clinical criteria AND is a contact of a probable or confirmed case, or linked to a COVID-19 cluster3, or a death linked to a cluster;

Confirmed: These are cases with a positive Nucleic Acid Amplification Test (NAAT), or a professional or self-test SARS-CoV-2 Antigen-RDT.

There are currently 194 member countries of the WHO, representing the full spectrum of social and economic development, and it is to be expected that the standard of diagnosis and reporting will vary considerably.

The range of symptoms is varied and is changing with the high rates of vaccination (in high-income countries) and the emergence of new variants and sub-variants. The increase in the availability of serological tests, Nucleic Acid Amplification Tests (NAAT) and especially the rapid antigen tests (RATS), means that the ‘Confirmed’ category is becoming a greater proportion of the reported cases. 

There is increasing recognition of the persistence of often debilitating symptoms after an acute COVID-19 infection which can last more than four weeks or even months after the initial infection. Sometimes the symptoms can even go away or come back again. Post-COVID-19 illnesses may present with a varied range and combination of symptoms with no fixed time frame. They may last for weeks or months and have varying degrees of disability. Most patients recover with time. 

### 1.2. Definitions of Long COVID-19

A post-COVID-19 condition occurs in individuals with a history of probable or confirmed SARS-CoV-2 infection, usually 3 months from the onset of COVID-19 with symptoms that last for at least 2 months and cannot be explained by an alternative diagnosis. Common symptoms include fatigue, shortness of breath, cognitive dysfunction and others that have an impact on everyday functioning. Symptoms may be new in onset following initial recovery from an acute COVID-19 episode or persist from the initial illness. Symptoms may also fluctuate or relapse over time [8]. The Australian Department of Health defines Long COVID-19 in the following way “where symptoms of COVID-19 remain, or develop, long after the initial infection—usually after 4 weeks. Symptoms of long COVID-19 can last for weeks or sometimes months” [9].

As the incidence of acute COVID-19 is declining, Long COVID-19 is increasingly being seen as an emerging public health problem. The symptoms overlap with chronic fatigue syndrome, myalgic-encephalomyelitis and other post-viral syndromes. Extensive research is underway to better define the diagnosis of Long COVID-19 and develop specific diagnostic tests. 

The US CDC states that people who experience post-COVID-19 conditions most commonly report the following symptoms [10]. General symptoms include fatigue that interferes with daily life, post-exertional malaise and unexplained fever. Respiratory and heart symptoms include dyspnoea, cough, chest pain and palpitations. Neurological symptoms are reported to include difficulty thinking or concentrating (sometimes referred to as “brain fog”), headache, insomnia and dizziness. paraesthesia, anosmia and depression or anxiety. Diarrhea and abdominal pain may continue along with joint or muscle pain and skin rashes. Changes in menstrual cycles have also been reported. 

Beyond the clinical symptoms, COVID-19 has had a huge impact on human populations, with effects well beyond mortality and morbidity including major economic and social disruption. It is likely that long COVID-19 will be a major public health issue as the world recovers from the pandemic. Some of these will be discussed in this paper.

## 2. Aim

It is important to include gender and equity issues when studying both acute and chronic diseases as these may be associated with causation and/or with outcomes. The current COVID-19 pandemic is a severe acute infection, and in addition, a proportion of those infected, continue to experience “long COVID-19 syndrome” where morbidity persists for several months or longer. The aim of this paper is to review gender issues in both acute COVID-19 and its long-term sequelae, referred to generally, in this paper, as ‘Long COVID-19’, with emphasis on Asia. The aim was assessed by the authors for consistency with the SANRA guidelines for narrative reviews [11].

## 3. Methodology 

The COVID pandemic has led to a tsunami of publications about COVID-19. According to SCOPUS, there were 210,183 COVID-19-related publications with 720,801 authors to August 2021 [12]. The number of papers has increased to 395,322 by October 2022. A wide range of disciplines were represented by the authors who have published on COVID-19, with all of the 174 scientific subfields in SCOPUS represented. While this was an unprecedented concentration of scientific talent it also meant that many authors ventured into areas where they were not qualified. At the same time, research publications in areas not related to COVID-19 have declined. Gender differences in the authors were analysed by Abramo and colleagues in ten countries. By comparing the rate of prepublications they concluded that while there had been a decline in the public rate of non-COVID-19 papers of 10–15%, the decline was uniform for both genders [13]. 

Due to the huge volume of publications and the number of databases to be searched we undertook a narrative review of gender issues related to COVID-19 and, where available, information on the incidence in the Asia-Pacific Region. There are many difficulties related to compiling accurate statistics on COVID-19 and some have been described by the WHO as it seeks to strengthen global information systems [14]. The major challenges with COVID-19 data have included the enormous size of the global pandemic, the sudden and unexpected onset, the difficulty of disease definition, especially prior to the availability of laboratory diagnosis, and now the varying access to diagnostic testing. In many communities, there is a reluctance to undergo testing and report cases as it may result in isolation and an inability to work. As well as monitoring cases in the community additional information has been obtained using other methods including serology surveys, excess mortality, life expectancy and post-mortem examinations. The amount of under-reporting has been estimated by WHO and Aizenman [15]. Countries with more reliable public health reporting systems in the region include Australia, Japan, New Zealand, the Republic of Korea, Taiwan and the USA. With the exception of the USA, these countries are all ranked in the first quartile of excess mortality [15] We will report on the public health and clinical impacts of COVID as they relate to gender in the Asia-Pacific Region and in some instances globally. The literature search was restricted to English papers published since 2020 and listed on the Web of Science. Search terms were COVID-19, Gender, Public Health, Asia and Review. Official government and recognised public health aggregation websites were included in this narrative review.

## 4. Results and Discussion

### 4.1. COVID-19 Cases and Deaths

Data on the incidence of COVID-19 have been collected by national public health ministries and organisations and have been consolidated by the World Health Organisation and other independent programs. The WHO uses the term ‘confirmed cases’ (see above), and as of the end of October 2022, it has recorded 627 million cases and 6.6 million deaths [2]. Results that are almost identical are reported by the Johns Hopkins Coronavirus Resource Center [16]. Totals are regularly updated, but COVID-19 gender data are incomplete.

COVID-19 data disaggregated by gender are provided by the Sex and Gender COVID-19 Project a partnership of Global Health 50/50, the African Population and Health Research Center (APHRC), and the International Center for Research on Women (ICRW) and is funded by the Bill and Melinda Gates Foundation [17]. However, gender-disaggregated data are difficult to interpret due to the lag in data reporting and the lack of gender information in most of the data. The available data are shown in Table 1.

In the table above there are some discrepancies between the figures from Global Health and the WHO due to different reporting dates and the completeness of the data. Gender is often missing, or reported at a different time, from other COVID-19 data and the gender distribution data may reflect this. The data from Australia, Japan, Korea, New Zealand, Taiwan and the USA for cases and deaths are regarded as complete. 

In Table 1. in Asia, female cases are recorded as approximately 10% greater than males in the countries with more complete data, but the number of deaths is about 20% lower. The case fatality rate, the proportion of confirmed cases who died, was approximately 20% lower in females. 

As well as the large national databases a large number of small case series have been published and subsequently included in meta-analyses. A meta-analysis of 3.1 million cases from 100 studies found that there was no difference in gender in contracting a COVID-19 infection [19]. However, males had higher mortality (OR 1.39; 1.31, 1.47) from COVID-19 and were more likely to be admitted to ICU (2.84: 1.45,3.79). This study included three reports from Asia, all from China. A further meta-analysis of 41 studies, including 18 from China, found that male sex, older age, obesity, diabetes and chronic kidney disease were associated with higher rates of mortality [20]. Both meta-analyses used data from the early stages of the pandemic. A more recent review that added studies from 2021 to previous analyses confirmed that the male sex was associated with increased mortality, admission to ICU and other severe outcomes [21]. In a scoping review of COVID-19 and gender in China, Feng concluded that more research is needed on gender and such an important disease [22]. 

Global excess deaths associated with COVID-19, have been modelled by WHO. In the following Table 2, the data from selected Asian countries are shown.

The WHO has developed protocols for documenting COVID-19 cases, including asymptomatic infections and vaccination effectiveness through serology surveys [23]. The data are consolidated on the SeroTracker Dashboard [24,25]. A global seroprevalence survey included surveys from three Asian countries that found no gender difference in prevalence [26]. In the future, it is likely that serology surveys will be used more often to monitor the epidemic.

The hospitalisation rate varies considerably according to the availability, accessibility and affordability of hospital beds. The data available for COVID-19-related admissions in Asia were limited. It reflects local accessibility more than other factors.

### 4.2. Life Expectancy 

This is the best overall single statistical measure of health and is now becoming available for the COVID era. Life Expectancy changes since the onset of the COVID-19 pandemic have been documented in 29 countries [27]. Changes in estimated life expectancy in the past 2 years are attributed to the pandemic. Only a few countries for which data are available did not have declines in life expectancy in 2020, including Norway, Denmark, Finland (for females only), New Zealand and Australia. Since 2020 countries in Western Europe have shown a recovery of life expectancy losses, but Eastern Europe and the United States continue to have life expectancy deficits [27]. In the USA the decrease in life expectancy undoes all of the gains in the last 26 years [28]. The Schöley review was unable to include analyses from lower and middle-income countries due to data limitations, but early studies of the pandemic from India suggest that there will be a substantial downturn in life expectancy there [29,30]. At this time, data on life expectancy changes in the Asian region are limited, but the available data are shown in Table 3.

The data currently available show that the gap between females and males in life expectancy has increased during the COVID-19 pandemic as males have had higher death rates from COVID-19. The most severe decline in period life expectancy so far documented in the USA where it is estimated that it has declined to the levels seen in 1996. The last time such a large drop in life expectancy occurred was during the 1918 Influenza Epidemic.

### 4.3. Long COVID-19

In the absence of definitive diagnostic tests, the documentation of long COVID-19 prevalence by gender is difficult and is dependent on assessing symptoms and the level of daily functioning, usually 3 months after the onset of the acute disease. It has a substantial impact on communities because of the loss of productivity at all levels of the economy and in the family. The commonly used definition in general use is “the condition that occurs in individuals with a history of probable or confirmed SARS-CoV-2 infection, usually 3 months from the onset of COVID-19, with symptoms that last for at least 2 months and cannot be explained by an alternative diagnosis” [38].

Estimates of the prevalence of Long COVID-19 show considerable variation with a range in prevalence from 9% to 81% in different studies, which is not surprising given the lack of a definitive diagnostic test [39]. A meta-analysis of 50 studies including 1.7 million subjects found an incidence of Long COVID-19 of 43% (95%CI 39,46%), 54% in subjects who had been hospitalised [40]. In Asia, the prevalence was 51% (95%CI 37,65%), higher than in Europe and the USA. This study also found that the female sex and pre-existing asthma had a higher proportion of post-COVID-19 conditions. Other chronic conditions are also likely to predispose to long COVID-19 with rates. Aiyegbusi also found an increase in symptoms in females but noted the need for better controls and gender-matched studies [41]. In a prospective cohort from Italy, females were found to be associated with a higher risk of developing “long COVID-19”, together with older age and smoking [42].

Further reviews which do not provide regional data also found that the likelihood of having long COVID-19 syndrome was higher in females, and they had a higher prevalence of psychiatric (depression), ear, nose and throat, and musculoskeletal and respiratory symptoms [39]. Mental health symptoms are a common feature of long-COVID-19 syndromes, including depression, sleep problems and fatigue and have a higher prevalence in women [43]. A review by Maglietta found significant associations between long COVID-19and female sex with any symptoms (OR 1.52 95% CI 1.27, 1.82), with mental health symptoms (OR 1.67 95% CI 1.21.2.29)and with fatigue (OR 1.54, 95% CI 1.32–1.79); and acute disease severity with respiratory symptoms (OR 1.66, 95% CI 1.03–2.68) [44] Long COVID-19 is also associated with many pre-existing conditions, including chronic obstructive pulmonary disease, fibromyalgia, anxiety, and coeliac disease, in addition to risk factors such as obesity, tobacco smoking, being female, and socioeconomic deprivation [45,46].

Children can also experience post-COVID-19 symptoms, and there are some reports that the prevalence of psychological complications is higher in female adolescents. A small cohort study of children from more than 120 days after their COVID-19 infection from Italy found 42.6% being impaired by these symptoms during daily activities 120 days after COVID-19, and no gender differences [47]. A review of the impact of the pandemic on children documented substantial negative impacts on their mental health. As well as during acute infection, COVID-19 has effects through illness and absence of care and social isolation during school closures and community lockdowns. These events caused relatively high rates of depression, anxiety, stress and post-traumatic stress disorder (PTSD) and suicidal behaviour. Adolescents had more of the psychological impact compared to children and female adolescents were at higher risk of experiencing anxiety, depression, and stress [48]. 

Vaccination is probably protective against developing long COVID-19. A cohort study of 3000 subjects in the UK given two doses found that 9.8% (CI 8.1, 10.6%) developed symptoms of long COVID-19 after 120 days. compared to 14.5% (CI 13.4, 15.9%) in controls [49]. When a vaccine was administered after contracting COVID-19, a first vaccine dose was associated with an initial 12.8% decrease (95% CI 18.6, 6.6%) in the odds of developing long COVID-19 [50], although others have found different results and further studies are needed [51].

In summary, long COVID-19 is at present difficult to diagnose and quantify empirically and women have a 10–20% higher rate than men. Full vaccination levels offer some protection.

### 4.4. Additional Burdens on Women during the COVID-19 Pandemic

These include difficulty in accessing healthcare (formal and informal) for women and their families, education at home, child care, orphans, subsistence food production, violence, and forced early marriages. There have been many orphans with estimates of numbers ranging from 2 to more than 10 million [52,53]. The burden of caring for them will fall heavily on female relatives as the pandemic eases.

### 4.5. Psychological Issues 

There are numerous studies highlighting the psychological difficulties faced during the pandemic. In addition to the stresses imposed by illness COVID-19 has led to severe restrictions on normal lives with the requirements of isolation, quarantine, travel restrictions and the lack of labour restricting usual services. During times of community disasters, including disease outbreaks, the burden of support in the family and community work borne by women tends to increase. This can result in pressures on health due to their greater responsibilities. Depression has been the most commonly associated with the pandemic and a review from the Asia-Pacific Region identified fear of COVID-19 infection (13%), females, (12%) and deterioration of underlying medical conditions (8.3%) as the most common associations. The levels of anxiety were greater in women than men (OR 1.44: 1.37, 1.52) [54]. Anxiety levels increased in pregnant women in Asia during the pandemic. This finding is in line with previous findings on gender differences in psychopathology during the COVID-19 pandemic in Southeast Asia [55,56,57]. China pursued a ‘no-COVID policy’ which led to prolonged lockdowns resulting in high levels of stress and depression [58]. In India, Malaysia and Japan there was an increase in parenting stress levels, compared to what they remembered from before there were COVID-19-related restrictions and school closures [59].

Female adolescents in Asia were less likely to have information about COVID-19. During the first year of the pandemic, the physical and psychosocial statuses of female youth were more negatively affected than male peers [60].

### 4.6. Violence against Women 

An international review found women were more likely than men (OR 1·23) to report that gender-based violence had increased during the pandemic [61]. In Asia violence against women and sexual violence against women and men has increased during the epidemic, in some countries by as much as 40% [62]. This includes rape in quarantine centres, refugee camps and even in an ambulance en route to hospital [63,64] A case study from a district in Pakistan reported that in one month there were 399 murders of women, but only 25 cases were reported to police [65,66]. Gender-based violence, including sexual violence, is all too common in the Asia-Pacific Region and increased during the COVID-19 pandemic [67]. Child marriage has increased again during the COVID-19 pandemic after a decline in recent years [68,69]. In Korea, the number of suicides remained relatively constant, but the proportion among females increased significantly [70]. 

### 4.7. Breastfeeding

Breastfeeding is the most important nutrition factor in the first year of life and has an impact on infants and their mothers throughout the formative years and beyond. There were doubts expressed initially about the safety of vaccinating mothers while pregnant or breastfeeding and of the possibility of transmitting the virus in breast milk or the effects of COVID-19 on lactation. Vaccination and breastfeeding are now known to be effective and good public health practices [71]. Although breastfeeding is a female activity men have an encouraging and supporting role. For optimal breastfeeding adequate nutrition and water supplies are required and, in many communities, these have been compromised due to labour shortages. Breastfeeding has been more important than ever during the pandemic to the lives of infants. Breastfeeding is crucial to public health, achieving sustainable development guidelines and mitigating the effects of climate change which affects women more than men [72]. During the pandemic. there was a shortage of infant formula and mothers who had ceased breastfeeding found it difficult to safely feed their infants [73]. In Vietnam, mothers chose to continue to breastfeed for the benefit of their infants, themselves and society as a whole [74]. COVID-19 vaccines are also safe during pregnancy and are important for the health of the mother [75]. During global emergencies, including the COVID-19 pandemic, the supply of manufactured infant formula has sometimes been interrupted. This makes it difficult for infants no longer being breastfed to receive adequate nutrition. 

### 4.8. Food and Nutrition and the Pandemic 

The pandemic has resulted in severe food shortages in many parts of the world due to reduced labour availability, transport interruptions and the lack of supplies of essential inputs, such as fertiliser and energy. The disruption to energy supplies has resulted in delays in harvesting, processing and distribution to consumers. During periods of lockdown, there have been issues with food issues shortages and distribution to homes. This disruption has been compounded by the war in Ukraine which has disrupted exports from the food bowl of Europe and the effects of climate change [76]. Women have borne much of the extra burden of increasing subsistence agriculture production to provide sufficient food for the family. The world prevalence of undernourishment jumped from 8.0 to 9.3 percent from 2019 to 2020 and to 9.8 percent, with 800 million people hungry in 2021 [77]. Women are the most vulnerable when food shortages occur preferring to give whatever food is available to their children. The prevalence of anaemia in women is high, approximately one in three, particularly in poor rural areas and is particularly dangerous during pregnancy. “Almost 3.1 billion people could not afford a healthy diet in 2020. This is 112 million more than in 2019, reflecting the inflation in consumer food prices stemming from the economic impacts of the COVID-19 pandemic and the measures put in place to contain it” [77].

In a comprehensive review of agriculture systems in Asia, Dixon and colleagues noted that COVID-19 accentuated existing gender inequities. They recommended that future structural adjustments and programs would be needed to improve equitable development, particularly for gender outcomes [78]. Myanmar was the site of a case study of farmers and found that women worried more about the impact of COVID-19 and reported an increase in their household work [79]. 

### 4.9. Impact of COVID-19 on the Health Workforce

Many Health workers have been the victims of the disease that they were working hard to help others overcome. The majority of cases were in females, but males have had most deaths (Bandyopadhyay 2022). The impact on elderly healthcare workers and those specialties more likely to be exposed to nasopharyngeal secretions is the most severe. In many countries, the political demand is to reduce pandemic restrictions, including wearing masks. As COVID-19 infections are still occurring health workers should continue to observe infection control measures as well as maintain their own full vaccination. The high rate of COVID-19 infection among health workers suggests that in the medium term the advent of long COVID-19 will severely impact the workforce. 

Health Care Providers are at increased risk during the pandemic due to high workload, high rate of illness amongst colleagues and the stress of dealing with a new and relatively unknown infectious disease. Females and nurses were at increased risk and a recent systematic review including 148 studies with 159,194 HCPs reported the pooled prevalence for depression was 37.5% (95%CI: 33.8–41.3), anxiety 39.7 (95%CI: 34.3–45.1), stress 36.4% (95%CI: 23.2–49.7), fear 71.3% (95%CI: 54.6–88.0), burnout 68.3% (95%CI: 54.0–82.5), and low resilience was 16.1% (95%CI: 12.8–19.4) [80]. 

The health workforce has a majority of women, in common with the other caring professions. They have experienced high rates of stress, anxiety and depression due to the pandemic and the need to work longer hours due to the workload and the loss of fellow workers to disease and burnout. An umbrella review of 40 systematic reviews (1,828 primary studies and 3.3 million subjects) reported high rates of mental stress in healthcare workers across the globe [81]. The most prevalent mental health problems identified in this review included anxiety, depression and stress, including PTSD. Other important mental health problems reported include burnout, insomnia, fear of infection, obsessive–compulsive disorder, phobia, somatisation symptoms, substance abuse, and suicidal ideation/self-harm. Significant risk factors associated with the incidence of mental health issues include female gender, young age, low educational level, being a nurse, being a frontline health professional, experience, and country of residence [81]. Women were at higher risk for several conditions including insomnia and depression and greater numbers in the health workforce and imposed a heavy load on therapists. 

Health workers have also had to reduce or abandon routine preventive healthcare including essential vaccinations. Resources have also been diverted into COVID-19 vaccination programs. Routine child vaccination programs were disrupted and 80 million children under 1 year missed routine vaccinations [82]. This will have an effect on future health and healthcare and will impact the achievement of the Sustainable Development Goal (3.8) of providing Universal Health Coverage by 2030.

### 4.10. National Workforce

The workforce during the pandemic has been decimated due to direct effects including deaths, infections and disability from Long COVID-19. Indirect effects include the impact of lockdowns and isolation measures. Some industries, including transport and education, have seen massive slowdowns. Many workers chose to work remotely rather than commute to an office, further reducing employment in the service industries. Childcare services were frequently closed or reduced and women often had additional home duties added when working from their homes. In Korea, women tended to take on the main responsibility for childcare, as usually, care facilities were not functioning, or relatives were not available due to isolation or illness [83]. A review of global statistics found that women were more likely to report employment loss (26·0% CI 23·8, 28·8) than men (20·4% CI 18·2, 22·9) during the first 18 months of the pandemic [61]. In Asia, more women than men increased their home duties and decreased paid employment and 80% of Asians reported a decrease in income [84]. A global online survey of professionals (*n* = 921) in biomedical fields working from home received responses from 76 countries. The change in work location was more likely to have a negative effect on women than on men. They felt more stressed by work expectations resulting in increased levels of anxiety and depression. Working from home was a challenge to women due to the lack of dedicated workspace and closeness to children and other family duties. However, the lack of travel and closeness to children was also a benefit. Being at home meant there was less interaction with friends and with other activities to relieve stress and depression [85]. 

### 4.11. Education

Schools around the world were closed during the pandemic and about one-third of children (430 million) did not have access to online education [86]. Education of girls is instrumental in improving child health and nutrition and in this way the pandemic may have a long-lasting effect on equality goals and health [87]. An estimated 11–20 million girls who were excluded from school during the pandemic will not return to school after the pandemic [63]. Reasons for this include sexual and gender-based violence, unintended pregnancy, forced marriage and employment. Where there is a shortage of teachers during and after the epidemic, girls are most likely to be discriminated against. Girls were more likely to drop out of school during the pandemic for reasons other than school closures, including the need to care for family members or to be employed [61]. In Asia even if education was still available, children from lower-income families were often forced to leave school as they could not afford the tuition fees [84]. During the pandemic, UNICEF estimated that 463 million children did not have access to remote learning and missed several years of education. 

Refugees and undocumented migrants have been victims of the pandemic, like everyone else, but have lacked treatment and social support services. There are at least 1.1 million Rohingya refugees from Myanmar living in the Cox’s Bazaar area of Bangladesh, 60% of children. Nutrition and health status are poor with a high rate of maternal and neonatal deaths [88]. There has been no vaccination program and no treatment available resulting in high rates of infection and death. This is typical of refugee situations in the region [89,90]. 

### 4.12. Science and Vaccination Developments

Justifiably the pandemic resulted in many studies and publications and the mobilisation of vast scientific resources to describe the virus and its variants and to develop successful vaccines. While scientific research has been intense, there has been far less success in ensuring that the benefits of research are extended to all of humanity. Vaccine access has been influenced by: country of residence, the wealth of the country and/or access to international donors, health infrastructure and the cold chain, location (rural/urban), socioeconomic grouping, education, gender and the influence of anti-vaccine propaganda.

Anti-vaxxers have been active since the beginning of modern medicine and the first introduction of variolation and later vaccination. A spate of cartoons lampooning smallpox variolation appeared at the beginning of the 19th century, most notably the famous cartoon by James Gilray in 1802 [91]. Misinformation about epidemics has of course been common throughout history. However, for the first time the availability of almost universal social media has allowed for the rapid spreading of news, but also nonsense pedalled by anti-vaxxers and other rumour mongers. In an online study from 10 countries, a majority from Asia (*n* = 1849) found that women were more likely to believe the science about the COVID-19 vaccine than men [92]. While some studies in single Asian countries have shown similar results with more women than men approving of vaccine science most have found no differences. By September 2021, women and men did not differ significantly in vaccine hesitancy or uptake [61].

Almost a million deaths have been averted by the use of COVID-19 vaccination and as a result, the number of lives saved by COVID-19 vaccination markedly exceeded the death toll that has occurred. Nonetheless, even more lives could have been saved by ensuring equitable worldwide vaccination coverage [4,93]. Despite early promises of worldwide distribution, there have been low rates of vaccination in low-income countries. Rates of vaccination in the Asian region vary from above 90% in some high-income countries down to below 10% as in Papua New Guinea where only 475,000 doses of vaccine have been administered to a population of 9.3 million [94].

### 4.13. Limitations

This is a narrative review and as an assessment of the COVID-19 pandemic, it is available earlier than a systematic review or meta-analysis. This review was restricted to the effects of gender COVID-19 and did not specifically include other co-variants that may be adjusted for in future studies as data become available. This may include variables related to ethnicity, healthcare access, education, socioeconomic status, Nutritional status and other biomedical variables. This study focused on gender issues in Asia and the results may not apply to citizens of other continents.

## 5. Conclusions

The COVID-19 pandemic has resulted in different impacts on females and males. The presently available data suggest that the rate of reported cases is about 10% greater in females than in males in Asian countries with more complete data. However, the reverse is the case for deaths, as males die more often than females from COVID-19. Rates of hospitalisation are more variable and are often determined more by access. The vaccination rates are approximately the same for both genders, and the overall rate is associated with the socioeconomic level of the country. 

Long COVID-19 will be a continuing public health burden, although the proportion of COVID-19 who will have longer symptoms is uncertain. Most estimates are that it is likely to be about 20%. Women experience the psychological consequences of COVID-19 more often than men. They also bear a greater proportion of the burden that the social and economic consequences of the pandemic on our societies. These factors will impose a continuing public health burden on society, regardless of the future direction of the pandemic. 

More ongoing research is required into COVID-19. Until now, research has rightly been concentrated on acute disease and prevention. Newer and more effective vaccines will need to be developed to combat emerging variants. More research is needed on long COVID-19 and in particular how to address its effects, especially in women and in Asia and lower socioeconomic regions.

## Figures and Tables

**Table 1 ijerph-20-00245-t001:** COVID-19 Cases and Deaths by Gender ASIA (Selected Countries).

**Country**	**Date (Mth Year)**	**Gender ASIA Report**								**WHO Report**	
		Cases			Deaths			% Died		Cases	Deaths
		Total	Male %	Female %	Total	Male %	Female %	Male	Female		
Australia	9/22	9,200,258	47.82	52.18	23,603	55.46	44.54	cc	0.2	10,332,884	14,853
Bangladesh	12/21	1,578,227	71	29	28,016	77	23	1.9	1.4	2,034,968	29,418
Cambodia	3/21	15,351	42.56	57.44	106			0.9	0.5	137,979	3056
China	8/22	315,059	53.74	46.25	2109	63.68	36.32	4.7	2.8	8,804,745	29,061
India	5/21	24,766,088	61.14	38.86				2.6	3.1	44,649,088	528,999
Indonesia	9/22	10,710,139	49.12	50.88	147,613	52.19	47.81	2.9	2.6	6,484,764	158,544
Japan	4/22	12,679,398	46.16	53.84	1,270,760	57.86	42.14	0.4	0.3	22,106	46,414
Korea Rep	6/22	18,248,479	47.01	52.99	24,399	48.71	51.29	0.1	0.1	25,431,105	29,069
Malaysia	9/22	9,321,616	52.12	47.88	71,700	57.02	42.98	0.8	0.7	4,887,675	36,458
Myanmar	10/21	139,349	50.02	49.98	2835	59.29	40.	2.6	1.8	631,235	19,480
Nepal	5/22	1,930,762	58.03	41.97	23,577	65.42	34.58	1.4	1.0	1,000,538	12,019
New Zealand	6/22	1,253,352	46.73	53.27	1294	52.86	47.14	0.1	0.1	1,811,552	3170
Pakistan	8/22	654,037	69.5	30.5	6190	74.15	25.85	2.1	2.1	1,573,725	30,624
Philippines	8/22	7,622,852	48.82	51.18	120,448	55.11	44.89	1.8	1,4	3,997,941	63,846
Sri Lanka	11/20	3088	81.64	18.36	74			1.8	5.1	671,037	16,777
Taiwan		3,189,338	47.77	52.33	11	81.8	18.2	1.6	0.4		
Thailand	11/20	3784	56.73	43.63	93	75.27	24.73	2.1	0.8	4,689,897	32,922
USA	6/22	75,463,792	46.78	53.24	845,385	55.06	44.94	1.4	1.0	95,946,824	1,059,255
Vietnam	11/21	9,872,600	45.2	54.8	86	43	57	0.4	0.4	11,498,873	43,162

% Deaths = deaths among confirmed cases. Proportion of confirmed cases that have died. Sources: [2,17], The Asia-Pacific Gender COVID report [18].

**Table 2 ijerph-20-00245-t002:** COVID-19 Excess Deaths by Gender in selected countries of ASIA.

Country	Date	Mean	Lower	Upper
Australia	12/21	−14,255	−10,510	−18,123
Bangladesh	12/21	140,764	49,648	237,705
Cambodia	12/21	12,518	1069	23,845
China	12/21	−52,063	-68,658	−35,996
India	12/21	4,740,894	3,308,100	6,479,698
Indonesia	12/21	1,028,565	750,749	1,286,532
Japan	12/21	−19,471	−35,732	−4245
Korea	12/21	6288	1440	11,254
Lao PDR	12/21	1756	−2956	6850
Malaysia	12/21	7533	−2537	18,220
Myanmar	12/21	44,188	200	87,815
Nepal	12/21	32,513	12,547	54,054
New Zealand	12/21	−2677	−3178	−2206
Pakistan	12/21	230,440	43,557	427,299
Philippines	12/21	185,253	165,274	208,241
Singapore	12/21	1475	1028	1927
Sri Lanka	12/21	−8831	−23,896	6461
Thailand	12/21	15,301	929	29,983
USA	12/21	932,458	886,917	978,225
Vietnam	12/21	−6211	−193,814	222,426

Note that the excess deaths can be negative as deaths from other diseases decline as a result of social changes during the pandemic. Overall, an excess of 14.9 million deaths over the number expected was recorded in 2021-21. Countries in Asia with the greatest number of excess deaths were India, Indonesia, Pakistan, the Philippines and the United States of America (USA) [3].

**Table 3 ijerph-20-00245-t003:** Life Expectancy Changes in the COVID-19 era.

Country	Pre-COVID-19 (2019)			COVID-19 (2020)		
	All	Female	Male	All	Female	Male
Australia		85	80.9		85.3	81.2
India		72.1	69.5		69.8	67.5
Japan		87.45	81.41		87.74	81.64
Korea		86.3	80.3		86.5	80.5
Singapore		85.9	81.4		85.9 (2021)	81.1 (2021)
Taiwan	80.9	84.2	77.7	80.86	84.25 (2021)	77.67 (2021)
USA	78.8	81.4	76.3	76.1 (2021)	79.1 (2021)	73.2 (2021)
Vietnam	73.74	78.11	69.56	73.6 (2021)	76.4 (2021)	71.1 (2021)

Sources: Australia [31], Japan [32], Korea [33], Singapore [34], Taiwan [35], USA [28,36] and Vietnam [37].

## Data Availability

This is a review article and the references have been carefully checked.

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
