# Peer review of "COVID and Gender: A Narrative Review of the Asia-Pacific Region"

_ijerph, 2022, doi:10.3390/ijerph20010245_

Round 1
Reviewer 1 Report
The paper presents some important results. The reserach methods are sound and the authors used appropriate resourses.
The authors need to improve their punctuation and tackle some grammatical problems.
Moreover, at times the writers use generalization. I pointed a few on the manuscript.
It would be interesting to see more data about the state of orphans and sinle mothers. These were mentioned very briefly in the article.
In conclusion, the article presents some interesting results and hence I advise to accept it after the recommendations above and on the manuscript are addressed.

Author Response
The paper presents some important results. The reserach methods are sound and the authors used appropriate resourses.
Thankyou for your careful review
The authors need to improve their punctuation and tackle some grammatical problems.
We have carefully reviewed the text and made numerous grammatical corrections
Moreover, at times the writers use generalization. I pointed a few on the manuscript.
Where data is available we have done our bet to be specific
It would be interesting to see more data about the state of orphans and sinle mothers. These were mentioned very briefly in the article.
We agree that this is an important social and public health impact of the pandemic. Unfortunately we have been unable to locate any more accurate and recent data
In conclusion, the article presents some interesting results and hence I advise to accept it after the recommendations above and on the manuscript are addressed.
Thankyou. We believe we have addressed all of the queries within the limits of currently available data.
Colin Binns
On behalf of all authors

Reviewer 2 Report
Introduction:
The introduction is well written. It comprehensively described the impact of COVID-19, highlighting the extrapolated numbers of death, definition and clinical symptoms of the disease. However, I felt some points were rather redundant. Would suggest the authors to consider adding more points on why focuses be given to "gender" and the population among "Asia".
Aim: -
Methodology:
Suggest the authors to rewrite the methodology section according to the SANRA or RAMESES guideline.
Certain points should be highlighted, i.e.: the search strategies, selection criteria before eventually bringing the readers to the results/discussion section.
Results:
There were certain non-Asia countries included in Table 2, would suggest rephrasing the title of Table 2.
Discussion:
The authors had appraised and discussed certain points and themes regarding gender and COVID-19, though some points may lackluster. Would like to suggest the authors to add on relevant appraisal in terms of the cultural background, education, childhood upbringing, socioeconomic status, religion, country counter-action policies, neuroendocrine/bio-chemical status among the diverse Asian females in comparison to males from Asia countries and females coming from the non-Asia counterparts.
Author Response
Thankyou for the considerable effort put into your review
Introduction:
The introduction is well written. It comprehensively described the impact of COVID-19, highlighting the extrapolated numbers of death, definition and clinical symptoms of the disease. However, I felt some points were rather redundant. Would suggest the authors to consider adding more points on why focuses be given to "gender" and the population among "Asia".
There have been no previous reviews of gender and COVID-19 in Asia. This article is part of the special issue on COVID and Gender. All of the authors live and work in Asia.
Aim: -
Methodology:
Suggest the authors to rewrite the methodology section according to the SANRA or RAMESES guideline.
The SANRA guidelines have been reviewed by the authors and addressed and referenced in the paper.
"It is important to include gender and equity issues when studying both acute and chronic disease as these may be associated with causation and/or with outcomes. The current COVID-19 pandemic is a severe acute infection and in addition a proportion of those infected, continue to experience “long COVID-19 syndrome” where morbidity persists for several months or longer. The aim of this paper is to review gender issues in both acute COVID-19 and its long-term sequelae, referred to generally, in this paper, as ‘Long COVID-19’, with emphasis on Asia. The aim was assessed by the authors for consistency with the SANRA guidelines for narrative reviews[11]."
Certain points should be highlighted, i.e.: the search strategies, selection criteria before eventually bringing the readers to the results/discussion section.
The research methodology section has been revised and expanded.
"We will report on the public health and clinical impacts of COVID as they relate to gender in the Asia Pacific Region and in some instances globally. The literature search was restricted to English papers published since 2020 and listed in Web of Science. Search terms were COVID19, Gender, Public Health, Asia and Review. Official government and recognised public health aggregation websites were included in this narrative review."
Results:
There were certain non-Asia countries included in Table 2, would suggest rephrasing the title of Table 2.
All of the countries mentioned are members of the Asia Pacific Academic Consortium for Public Health and annually participate in Asian Health forums. They are also a part of various regional economic organisations.
Discussion:
The authors had appraised and discussed certain points and themes regarding gender and COVID-19, though some points may lackluster. Would like to suggest the authors to add on relevant appraisal in terms of the cultural background, education, childhood upbringing, socioeconomic status, religion, country counter-action policies, neuroendocrine/bio-chemical status among the diverse Asian females in comparison to males from Asia countries and females coming from the non-Asia counterparts.
This paper concentrates on gender. For a systematic review or meta-analysis a large number of covariates could be included. Unfortunately this data is not yet available.
In response to the review a new Limitations section has been added which includes these factors.
"
Limitations
This is a narrative review and as an assessment of the COVID-19 pandemic it is available earlier than a systematic review or meta-analysis. This review was restricted to the effects of gender Covid-19 and id not specifically include other co-variants that may be adjusted for in future studies as data becomes available. This may include variables related to ethnicity, health care access, education, socioeconomic status, Nutritional status and other biomedical variables. This study focused on gender issues in Asia and the results may not apply to citizens of other continents."
We have made grammatical and spelling corrections to the text.
Thank you for your helpful suggestions and we believe that we have addressed them all.
Colin Binns
On behalf of the authors